# Morphological and Molecular Characterization of *Calonectria foliicola* Associated with Leaf Blight on Rubber Tree (*Hevea brasiliensis*) in Thailand

**DOI:** 10.3390/jof8100986

**Published:** 2022-09-20

**Authors:** Narit Thaochan, Chaninun Pornsuriya, Thanunchanok Chairin, Putarak Chomnunti, Anurag Sunpapao

**Affiliations:** 1Agricultural Innovation and Management Division (Pest Management), Faculty of Natural Resources, Prince of Songkla University, Hatyai 90110, Thailand; 2School of Science, Mae Fah Luang University, Chiang Rai 57100, Thailand

**Keywords:** emerging disease, leaf disease, morphology, molecular identification, pathogenicity

## Abstract

Leaf blight is commonly observed in rubber trees (*Hevea brasiliensis*) and can be caused by several fungal species. From October to December 2021, the emergence rubber tree disease was observed in Krabi province, southern Thailand. Small brown to dark brown spots developed on the leaves of rubber trees and later expanded into most parts of the leaves. Fungal isolates were isolated from infected tissues and a total of 15 *Calonectria*-like isolates were recovered from 10 infected leaf samples. Pathogenicity testing using the agar plug method revealed that four isolates caused leaf blight on rubber tree, similar to the situation in natural infections. Based on morphological study and the molecular properties of internal transcribed spacer (ITS), calmodulin (*cal*), translation elongation factor 1-α (*tef1-α*), and β-tubulin 2 (*tub2*) sequences, the four fungal isolates were identified as *Calonectria foliicola*. To the best of our knowledge, this is the first report of rubber trees pas a new host for *C*. *foliicola* in Thailand and elsewhere. This study reports on an emerging disease affecting rubber trees in Thailand, and the results are of benefit for the development of an appropriate method to manage this emerging disease in Thailand.

## 1. Introduction

Rubber tree or Para rubber tree is an angiosperm plant belonging to the Euphorbiaceae family and that is native to the Amazon rainforest [1,2]. Among the various plant species, *Hevea brasiliensis* is economically important due to its capacity to produce milky latex as the source of natural rubber, with good yield and excellent physical properties [3,4,5] and suitable for use in numerous applications [6,7]. This plant species was introduced to Asia at the end of 19th century [8]. Currently, the major plantation areas for rubber trees are found in Southeast Asia, especially in Indonesia, Malaysia, Thailand, and Vietnam, where they represent more than 1.5 million ha [9].

The cultivation of rubber trees is hampered by biotic stress, especially due to plant pathogenic fungi and fungi-like organisms that can cause numerous diseases in all stages of growth. For instance, leaf spot disease is caused by *Corynespora cassiicola* [10] and *Neopestalotiopsis aotaroa* [11]. Anthracnose caused by *Colletotrichum siamense* and *C. australiensis* has been recorded as a major causative agent in China [12]. The fungus *Chalaropsis thielavioides* is reported as a pathogen for wilt in rubber trees in China [13]. *Bipolaris bicolor* is found to cause leaf spots in rubber trees [14]. In Thailand, fungal pathogens and fungal-like organisms have been reported to cause diseases in rubber tree, especially in southern Thailand, where the weather conditions favor pathogen germination and disease spread [15,16]. Algal leaf spot caused by *Cephaleuros virescens* is commonly observed in hot and humid regions in southern Thailand [17]. *Phytophthora citrophthora* causes leaf fall disease in southern Thailand [18]. Furthermore, the cause of an emerging leaf fall disease of rubber trees in this area has recently been reported as fungi *Neopestalotiopsis cubana* and *N. formicarum* [19].

Leaf disease is one of the major diseases of rubber trees that can be caused by diverse fungi [7,13,14,20,21]. *Calonectria* is a fungal genus in the family Nectriaceae, whose species inhabit soil [22,23] and are plant pathogens [24,25,26]. From October to December 2021, the occurrence of emerging leaf blight disease was observed in Krabi province, southern Thailand. However, the identity of the fungi associated with leaf blight in this area is unclear. Therefore, this research aimed to identify the fungal pathogen responsible for causing leaf blight disease of rubber tree using a combination of morphology and molecular tools as well as to test pathogenicity according to fulfillment of Koch’s postulates.

## 2. Materials and Methods

### 2.1. Sample Collection

Leaf spots on rubber trees were collected from Krabi province, southern Thailand, from October to December 2021. A total of 10 symptomatic *Calonectria*-like leaf samples were collected in plastic bags, kept in a cooler box, and brought to the Plant Pathology Laboratory, where isolation was subsequently conducted. Fungi were isolated from symptomatic leaves using a tissue transplanting method [27,28] with some modifications. Small pieces (3 mm × 3 mm) of symptomatic tissues containing healthy tissues were cut using a razor blade, surface disinfected using 0.5% sodium hypochlorite (NaOCl) [29] and rinsed with sterile distilled water (DW). The samples were hung to dry on sterile Whatman membranes and then placed on water agar (WA). The PDA plates were incubated at ambient temperature (28 ± 2 °C) for 2 days. Hyphal tips recovered from tissue samples were cut and transferred to potato dextrose agar (PDA) for further study.

### 2.2. Pathogenicity Test

To test whether fungal isolates can cause disease in rubber tree leaves, pathogenicity testing was conducted according to the agar plug (0.5 mm) method [30,31] to determine fulfillment of Koch’s postulates. Healthy rubber tree leaves were disinfected using 70% ethanol. There were 4 treatments, including control leaves (wounded and unwounded) inoculated with PDA, and leaves inoculated with each fungal isolate. Agar plugs of fungal isolates were cut from the edge of a 5-day-old colony and placed on rubber tree leaves. Inoculated leaves were incubated in a moist box at ambient temperature (28 ± 2 °C). There were 3 leaves per treatment and the experiment was repeated twice. Samples were observed daily for symptom development. Fungi were re-isolated from inoculated symptomatic leaf samples to confirm correspondence with the same pathogen used for inoculation based on morphological observation.

### 2.3. Molecular Study

The selected fungal isolates from Section 2.2 were cultured on PDA for 2 days to obtain young mycelia, which were subjected to DNA extraction using a mini-preparation method [32]. The presence of total DNA was observed by 1% agarose gel electrophoresis. PCR amplification was performed in a thermal cycler (Bio-Rad Laboratories, CA, USA). Internal transcribed spacer (ITS), calmodulin (*cal*), translation elongation factor 1-α (*tef1-α*), and β-tubulin 2 (*tub*2) were amplified using ITS5/ITS4 [33,34], CAL-228F/CAL-737R [35], EF1-728F/EF2 [35,36], and T1/Bt-2b [33,36] primer pairs, respectively. The PCR mixture was composed of 2 μL DNA template, 20 *p*mol of each primer, 2× OneTaq^®^ PCR master mix with standard buffer (Biolabs, New England, MA, USA), and nuclease-free DW. PCR amplifications were performed with an initial denaturation step at 94 °C for 30 s followed by 30 cycles of denaturation at 94 °C for 30 s, annealing at 60 °C for 60 s, and extension at 68 °C for 1 min, and final extension at 68 °C for 5 min. PCR products were stained with Novel Juice (GeneDirex, Taoyuan, Taiwan) and visualized using 1% agarose gel electrophoresis in 0.5× TBE buffer.

PCR products were sequenced at Macrogen (Seoul, Republic of Korea). The DNA sequences of ITS, *cal*, *tef1-α*, and *tub2* were searched in databases using BLASTn (National Center for Biotechnology Information, NCBI). The DNA sequences of fungal isolates and related species were acquired from the GenBank database (Appendix A) to construct the phylogenetic tree analysis. DNA sequences of ITS, *cal*, *tef1-α*, and *tub2* were deposited in GenBank with accession numbers.

Sequences were aligned with Bioedit v. 7.2 [37] using a ClustalW algorithm and ClustalX v. 1.83 [38] and manually adjusted as necessary. Phylogenetic tree estimation for each alignment was performed using maximum likelihood (ML), maximum parsimony (MP), and Bayesian inference (BI). An ML tree was constructed using MEGA X based on nearest-neighbor-interchange (NNI) as the heuristic method for tree inference, and 1000 bootstrap replicates were performed. MP tree was obtained using the heuristic search option with 1000 random additions of sequences and tree bisection and reconnection (TBR) as the branch-swapping algorithm of MEGA X [39]. The Bayesian tree was generated using MrBayes ver. 3.2.7a [40]. Markov chain Monte Carlo (MCMC) runs were performed for 1,000,000 generations and sampled every 100th generation. The initial 25% of generations were discarded as burn-in, and the remaining trees were used to calculate the Bayesian inference posterior probability (BIPP) values. Phylogenetic trees were visualized using FigTree ver. 1.4.4 (http://tree.bio.ed.ac.uk/software/figtree/, accessed on 4 July 2022).

### 2.4. Morphological Study

The selected fungal isolates were cultured on PDA and incubated at ambient temperature for 7 days for morphological study. Colony growth and growth rate were measured. Macroscopic and microscopic features of fungal isolates were observed under a Leica S8AP0 stereomicroscope (Leica Microsystem, Wetzlar, Germany) and Leica DM750 compound microscope (Leica Microsystem, Wetzlar, Germany). The dimensions of fungal conidia were measured (*n* = 20).

### 2.5. Statistical Analysis

The lesion size of unwounded and wounded rubber tree leaf was subjected to one-way analysis of variance (ANOVA). The difference in lesion size was analyzed by Tukey’s HSD test (*p* < 0.05).

## 3. Results

### 3.1. Symptom Recognition and Fungal Isolates

The occurrence of leaf blight was observed in 3% of diseased leaf samples in this study. The symptoms start as circular brown spots that progress into having brown to dark brown crenate edges in radial or irregular expansion (Figure 1). In old leaves, the symptoms resembled soaked water and were observed in the middle of the affected area in cases of severe case leaf blight (Figure 1). A total of 15 fungal isolates, which were recovered from 10 infected tissues using a tissue transplanting method, were subjected to pathogenicity testing.

### 3.2. Pathogenicity of Fungi

In order to test whether fungal isolates recovered from infected tissue can cause disease, pathogenicity testing was conducted on rubber tree leaves. A total of 15 isolates were tested for their pathogenic on wounded and nonwounded rubber tree leaves, and only 4 isolates, namely Calkb001, Calkb002, Calkb003, and Calkb004, caused disease symptoms to appear on rubber tree leaves at 3 days post-inoculation (Figure 2). When comparing wounded and unwound leaves, rubber tree leaves inoculated by all isolates showed severe symptoms on wounded leaves (Table 1). Fungal isolates were re-isolated from inoculated leaves, and the results confirmed that the morphology was similar to that observed for the first isolation. Then, the 4 isolates were selected to molecular identification.

### 3.3. Molecular Identification

BLAST searching of GenBank (The National Center of Biological Information, NCBI) showed the sequences of ITS, *cal*, *tef1-α*, and *tub2* regions were more than 99% identical to those in *Calonectria foliicola* for all isolates examined in this study. The dataset comprises 45 taxa, including *Curvicladiella cignea* as the outgroup. It consists of 1956 characters including gap (ITS: 528, *cal:* 446, *tef1-α*: 464 and *tub*: 518). The most parsimonious tree generated from combined DNA sequences of ITS, *cal*, *tef1-α* and *tub2* of *C*. *cylindrospora* species complex is shown in Figure 3. The four isolates Calkb001, Calkb002, Calkb003, and Calkb004 form an independent clade with ML (62%), MP (76%), and BYPP (0.95) supports and were close to *C*. *foliicola*. DNA sequences of each fungal isolates were deposited in GenBank with accession numbers (Appendix A).

### 3.4. Morphology Identification

Four isolates (Calkb001, Calkb002, Calkb003, and Calkb004) were obtained from infected tissues of infected rubber tree leaves. All showed similar morphology and, therefore, the Calkb001 isolate is used as the basis for describing morphology in this study. All isolates showed similar growth rate 6.69 ± 0.01 mm/day on PDA (*n* = 6) with an irregular margin, producing abundant white aerial mycelia that were orange on the reverse side of the plate (Figure 4). Description: Teleomoph not found. Anamorph: Macroconidiophores consisting of a stipe, a suite of penicillate settled fertile branches, a stipe allowance, and a terminal vesicle; stipe septate, hyaline, smooth, 76–85 × 4–8 μm; stipe extension with septate, straight to flexuous 22–94 μm long, 1.5–2.5 μm wide at the apical septum, terminating in an obpyriform to ellipsoidal vesicle, 8.2–15.7 μm diam. Conidiogenous apparatus 31–70 μm long, 13–67 μm wide; primary branches without septate, 16–26 × 3.5–5.8 μm; secondary branches without septate, 11–20 × 3.4–5.0 μm; tertiary branches without septate, 8–11 × 2.7–4.8 μm; 2–4 phialides produce in each terminal branch; phialides doliiform to reniform, hyaline, without septate, 8–13 × 3.5–4.8 μm; apex with minute periclinal thickening and inconspicuous collarette. Macroconidia cylindrical, curved at both ends, straight, 44–48 × 3.5–4.3 μm (av. = 45.6 × 3.6 μm) with 1 septate. Mega- and micro-conidia were not found.

## 4. Discussion

Thailand is located in tropical and subtropical areas, and the weather favors pathogen germination and spread [15,16]. Identification and characterization of fungal pathogens in this area is most important as the first step of disease management. To date, only *Neopestalotiopsis* and *Phytophthora* are commonly reported to cause major disease in rubber trees [18,19]. With no prior reports related to the disease causing *Calonectria* species in Thailand, this research aimed to study the rare *Calonectria* species causing leaf blight in rubber trees. In this study, the pathogens causing the emerging disease or “leaf blight” of rubber trees were characterized according to both morphology and molecular properties of multiple DNA sequences as *C*. *folliicola* and subjected to pathogenicity testing.

The growth rate of fungal colony of *C*. *foliicola* observed in this study was identical to that observed in previous reports [41,42]. In the present study, all four isolates (Calkb001, Calkb002, Calkb003, and Calkb004) grew to cover the 9 cm Petri dish within 14 days. However, several reports showed that the colony shapes of *Calonectria* vary based on species and isolates [41,42]. Our *C*. *foliicola* (Calkb001, Calkb002, Calkb003, and Calkb004) displayed dimensions (shape and size) of macroconidiophores, conidiogenous apparatus, phialides, and macroconidia similar to earlier observations described in several reports [41,42,43]. Our results for Calkb001, Calkb002, Calkb003, and Calkb004 are in agreement with previous reports indicating their morphology is typical of *C*. *foliicola*.

In this study, pathogenicity testing revealed that the four isolates caused symptoms on rubber tree leaves. When comparing between unwounded and wounded, our results showed that wounding results in successful inoculation of *Calonectria* into rubber tissues. However, symptom development, though slower, was observed in unwounded rubber tree leaves. This phenomenon suggests that wounding may help fungal pathogens penetrate into plant tissues to colonize and cause infection, resulting in more rapid symptom development. This finding is in agreement with previous reports that wounding is necessary for the pathogenicity of several fungal pathogens [29,44]. Furthermore, brown rot tissue symptoms may also be associated with activities related to peroxidase and polyphenol oxidase in defense response [45]. Nevertheless, we did not characterize enzyme activity in this study. Based on the results in this study, *C*. *foliicola* causes leaf blight of rubber tree leaves.

Currently, the identification of fungal pathogens causing diseases in plants relies on both morphology and molecular techniques in addition to pathogenicity testing to determine whether Koch’s postulates are fulfilled [19,31,46]. The molecular details of multiple DNA sequences have been applied in identifying fungal species [42,43]. For instance, Bose et al. [47] used two DNA sequences of *tub2* and *tef1-α* to confirm the species of *C*. *cerciana* causing leaf blight of Eucalyptus in northern India. Stępniewska et al. [48] also used multiple DNA sequences of *tef1-α*, histone H3 (*his3*) and *tub* to identify *C*. *montana* causing damping-off disease on pine and spruce seedlings in Europe. For *C*. *foliicola*, this species was firstly described by Lombard et al. [43] as a new species by comparison of morphology and multiple DNA sequences of *tub*, calmodulin (*cal*), *his3*, and *tef1-α*. In this study, we used comparison of multiple DNA sequences of ITS, *cal*, *tef1-α*, and *tub2* and combined these sequences to construct phylogenetic trees (Figure 3), identifying the four fungal isolates as *C*. *foliicola*. In the present study, our results are in agreement with the previous reports mentioned above, with multiple nucleotide sequences of ITS, *cal*, *tef1-α*, and *tub2* successfully identifying Calkb001, Calkb002, Calkb003, and Calkb004 as *C*. *foliicola*.

The genus *Calonectria* displays *Cylindrocladium*-like asexual morphs that are more commonly found in nature than sexual morphs [49]. Currently, *Calonectria* spp. act as pathogens in causing extensive losses in forestry plantation in tropical and subtropical areas worldwide [50,51,52]. A total of 335 plant host species are susceptible to *Calonectria* spp. [53], in which numerous diseases are reported. For instance, *C. pteridis* has been reported to cause leaf spots on *Serenoa repens* in China [54]. *C*. *metrosideri* causes leaf blight on *E. benthamii* in Brazil [55], whereas *C. tunisiana* causes crown and root rot of *Eucalyptus globulus* in Italy [56]. However, *C. foliicola*, which belongs to the *C*. *cylindrospora* species complex, has only been reported in *E. grandis* and *E. urophylla* [42,43]. However, there are no reports of *C*. *foliicola* causing leaf blight disease on rubber trees in Thailand and elsewhere. This is the first report of rubber trees (*H. brasiliensis*) as a new host for *C. foliicola*.

## Figures and Tables

**Figure 1 jof-08-00986-f001:**
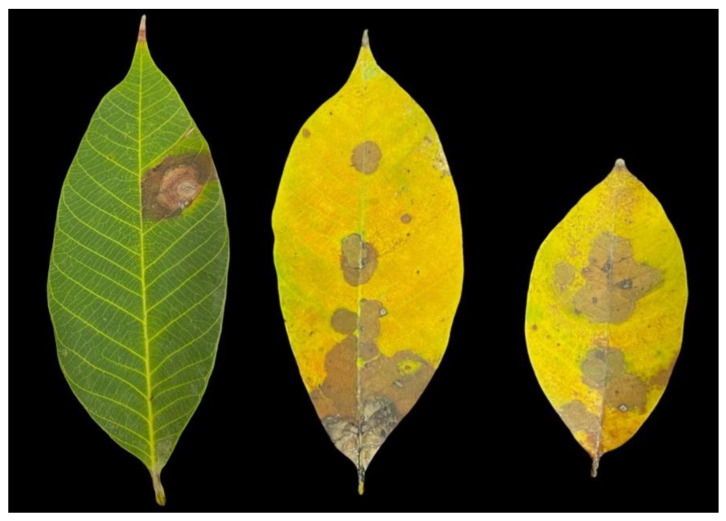
Disease symptoms of leaf blight disease on rubber trees.

**Figure 2 jof-08-00986-f002:**
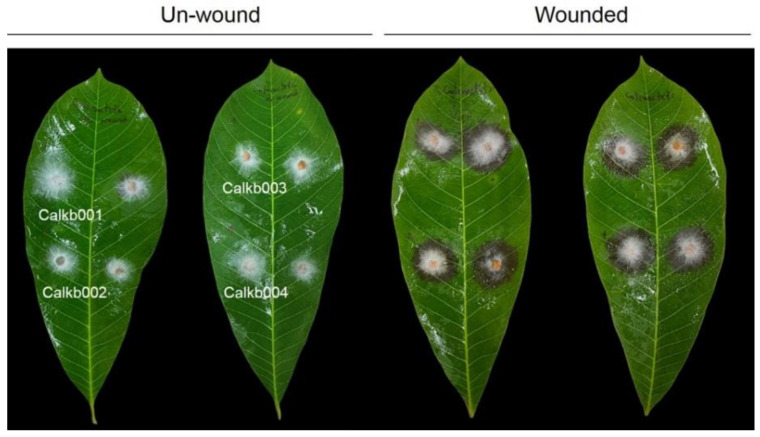
Pathogenicity test of leaf blight symptoms caused by the *Calonectria foliicola* isolates Calkb001, Calkb002, Calkb003, and Calkb004 on unwounded and wounded rubber tree leaf after 3 days post-inoculation.

**Figure 3 jof-08-00986-f003:**
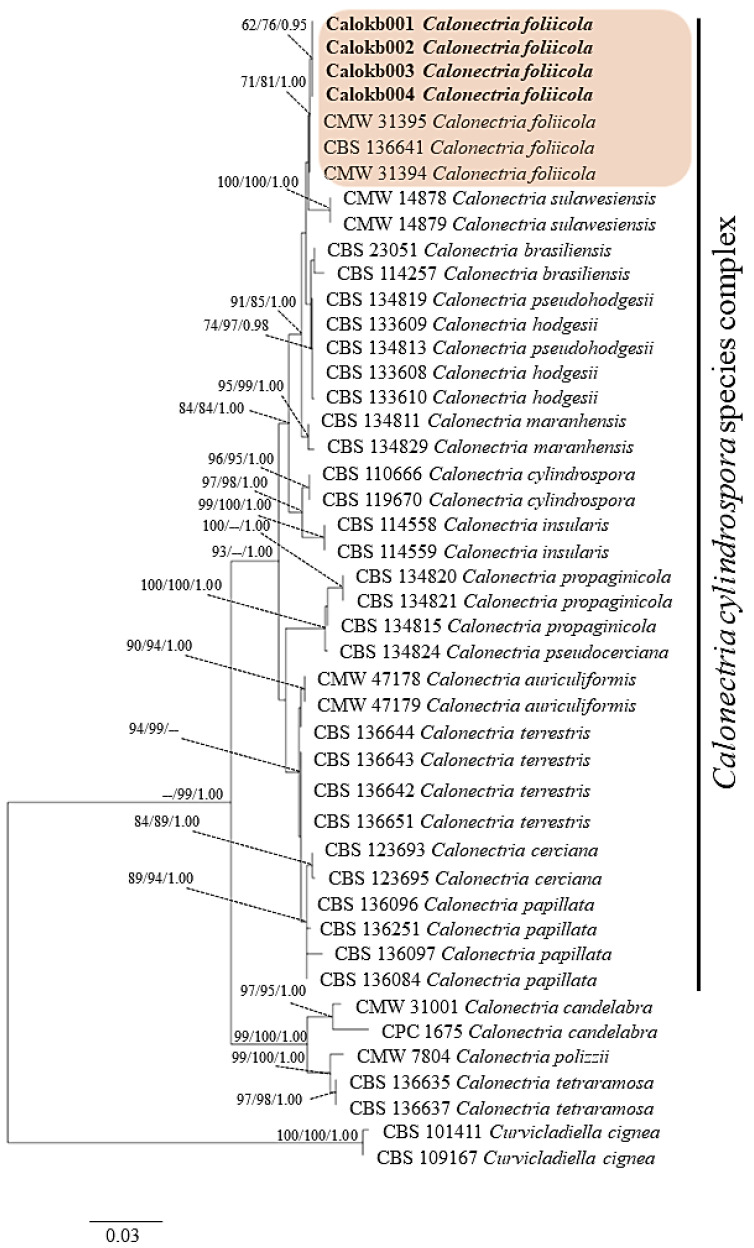
Phylogenetic tree generated from maximum likelihood (ML) and based on the combined sequences of ITS, *tub2, cal*, and *tef1-α* sequence data of *Calonectria* spp. in the *C**. cylindrospora* species complex. Bootstrap values for maximum likelihood (ML) and maximum parsimony (MP) equal to or greater than 50% and Bayesian posterior probabilities (BYPP) equal or greater than 0.95 are indicated. The isolates obtained in this study are indicated in bold. The tree is rooted to *Curvicladiella cignea*.

**Figure 4 jof-08-00986-f004:**
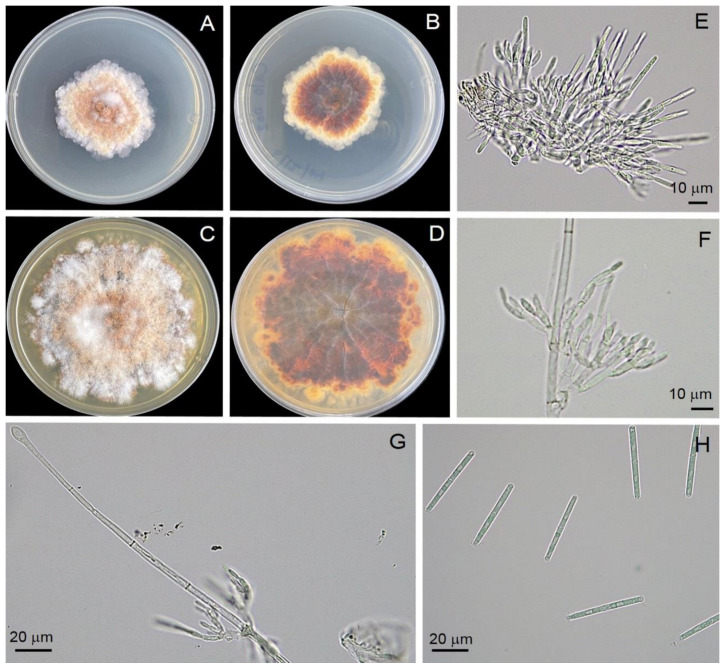
Morphological characteristics of the *Calonectria foliicola* Calkb001. Top view (**A**) and bottom view (**B**) of a colony on PDA at 7 days, top view (**C**) and bottom view (**D**) of a colony on PDA at 14 days, conidiogenous apparatus with conidiophore branches and doliiform to reniform phialides (**E**,**F**), obpyriform to ellipsoidal vesicle (**G**), macroconidia (**H**).

**Table 1 jof-08-00986-t001:** Comparison of lesion size (mm) following inoculation of *Calonectria* on rubber tree leaves, unwounded and wounded.

*Calonectria* Isolates	3 dpi. ^1^		5 dpi.	
	Unwounded ^2^	Wounded	Unwounded	Wounded
Calkb001	-	24.8 ± 2.3	17.0 ± 9.8	54.3 ± 1.5
Calkb002	-	24.3 ± 0.3	-	57.5 ± 0.3
Calkb003	-	21.5 ± 1.2	11.0 ± 8.5	51.0 ± 2.3
Calkb004	-	23.5 ± 1.3	-	56.3 ± 3.0
		ns ^3^		ns

^1^ day post-inoculation (dpi), ^2^ values are means ± SE. ^3^ not significantly different at *p* > 0.05 followed by Tukey’s HSD test.

## Data Availability

Not applicable.

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
