# Peer review of "Morphological and Molecular Characterization of *Calonectria foliicola* Associated with Leaf Blight on Rubber Tree (*Hevea brasiliensis*) in Thailand"

_jof, 2022, doi:10.3390/jof8100986_

Round 1

Reviewer 1 Report

This study, by Thaochan and colleagues, for the first time reports the association of the fungal pathogen Calonectria foliicola with leaf blight of rubber plants in Thailand and other parts of the world. The paper is written nicely. However, before being accepted for publication, I recommend authors clarify some of the issues that I have mentioned below. 

Line 14. Make it small brown to dark brown.

Line 76. Mention the size of the agar plug.

Line 86. Mention the buffer (with its strength) used for electrophoresis.

Line 91. Mention the amount of DNA used for PCR.

Line 98. It should be BLASTn.

Line 133. 15 isolates were recovered from 10 infected leaves which means some leaves were infected by more than one type of fungus? Please clarify this statement.

Line 138. Make it "for their pathogenicity" or "their virulence".

Line 141. Here, you mentioned all isolates, does this mean those four isolates that produced symptoms on rubber leaves, or all 15 isolates that were recovered from infected leaves? Please make it clear.

Line 154. So, all 15 isolates were members of Calonectria foliicola but only four caused symptoms on detached rubber leaf? If this is tru then please discuss why other 11 Calonectria foliicola isolates did not produce disease symptoms on detached leaves.

Line 190. “leaf blight”

Figure 3. Include DNA sequences for all 15 Calonectria foliicola isolates in the phylogenetic analysis

Table S1. I can see several isolates that were being used as references from GenBank lacking ITS sequences (not available). In that case, why those isolates were included in the multi-locus phylogenetic analysis (based on 1956 characters)?

Author Response

This study, by Thaochan and colleagues, for the first time reports the association of the fungal pathogen Calonectria foliicola with leaf blight of rubber plants in Thailand and other parts of the world. The paper is written nicely. However, before being accepted for publication, I recommend authors clarify some of the issues that I have mentioned below. 

Line 14. Make it small brown to dark brown.

Answer We have revised, see at line 14.

Line 76. Mention the size of the agar plug.

Answer We have revised, see at line 76

Line 86. Mention the buffer (with its strength) used for electrophoresis.

Answer We have revised, see at line 100.          

Line 91. Mention the amount of DNA used for PCR.

Answer We have revised, see at line 94.

Line 101. It should be BLASTn.

Answer We have revised, see at line 102.

Line 133. 15 isolates were recovered from 10 infected leaves which means some leaves were infected by more than one type of fungus? Please clarify this statement.

Answer One leaf have more than 1 wound, therefore, 1-2 fungal isolates could isolate from 1 infected leaf.

Line 138. Make it "for their pathogenicity" or "their virulence".

Answer for their pathogenicity see at line 139.

Line 141. Here, you mentioned all isolates, does this mean those four isolates that produced symptoms on rubber leaves, or all 15 isolates that were recovered from infected leaves? Please make it clear.

Answer Only 4 isolates showed the symptoms on rubber tree leaves at 3 days post inoculation.

Line 154. So, all 15 isolates were members of Calonectria foliicola but only four caused symptoms on detached rubber leaf? If this is true then please discuss why other 11 Calonectria foliicola isolates did not produce disease symptoms on detached leaves.

Answer Primary morphological character of 11 isolates similar to Calonectria sp. but did not cause the disease symptom on the rubber leaf. We selected only 4 isolates that showed the pathogenicity and virulence on the rubber leaf for further study.

Line 190. “leaf blight”

Answer Revised already see at line 200.

Figure 3. Include DNA sequences for all 15 Calonectria foliicola isolates in the phylogenetic analysis

Answer We selected only 4 isolates for molecular study that we mention at line 154.

Table S1. I can see several isolates that were being used as references from GenBank lacking ITS sequences (not available). In that case, why those isolates were included in the multi-locus phylogenetic analysis (based on 1956 characters)?

Answer: Because those isolates showed the most identical to Ca. foliicola found in this study, but their identification were constructed the phylogenetic tree without ITS sequences. Therefore, combination of tub2+cal+tef1-α sequences was sufficient to distinguish this species to others.

Reviewer 2 Report

Overall, I found out that:

1. The subject of the manuscript falls within the proper scope of the Journal and contains enough information/data for publication.

2. Methodology, analyses and approaches used to answer the research questions are well reasoned and appropriate; sufficient detail is provided.

3. I have a concern regarding the originality of your writing as I have found almost identical word arrangements from other articles in your Methodology and section Results; Morphology Identification. I would suggest you to-rewrite using your own words and not copy-pasting from other articles to avoid plagiarism.

Author Response

Overall, I found out that:

  1. The subject of the manuscript falls within the proper scope of the Journal and contains enough information/data for publication.
  2. Methodology, analyses and approaches used to answer the research questions are well reasoned and appropriate; sufficient detail is provided.
  3. I have a concern regarding the originality of your writing as I have found almost identical word arrangements from other articles in your Methodology and section Results; Morphology Identification. I would suggest you to-rewrite using your own words and not copy-pasting from other articles to avoid plagiarism.

Answer We have revised according to your comments.